# Estimation of the return period of rockfall blocks according to their size

Valerio De Biagi[1], Maria Lia Napoli[1], Monica Barbero[1], and Daniele Peila[2]

[1]Department of Structural, Geotechnical and Building Engineering, Politecnico di Torino, Torino, Italy
[2]Department of Environmental and Infrastructural Engineering, Politecnico di Torino, Torino, Italy

*Correspondence to:* Valerio De Biagi (valerio.debiagi@polito.it)

**Abstract.** With reference to the rockfall risk estimation and the planning of rockfall protection devices one of the most critical and most discussed problems is the correct definition of the design block taking into account its return period. In this paper, a methodology for the assessment of the design block linked with its return time is proposed and discussed, following a statistical approach. The procedure is based on the survey of the blocks already detached from the slope and accumulated at the foot of the slope and the available historical data.

## 1 Introduction

Rockfall is one of the most critical slope instabilities because it can be highly destructive and unpredictable. The analysis of this phenomenon is very difficult because it is affected by aleatory variability (irreducible natural variability) and epistemic uncertainty (lack of knowledge). For these reasons, probabilistic methods are the suitable approaches for modeling rockfall. When risk analysis has to be performed for forecasting and protection purposes, the size of the involved blocks and the corresponding return period are the most important variables among the ones that characterize the phenomenon (Peila et al., 1998, 2006; Peila and Ronco, 2009).

Modern design approaches for buildings, for example, aim at guaranteeing the structural safety throughout its expected life. In such reliability-based framework, the buildings have to be robust, i.e., to support forces due to anthropic and natural hazards without being significantly damaged. Proper design processes for common natural hazards, such as extreme winds or seisms, are already present in the building codes (Elishakoff, 1999; ISO, 1998; Leporati, 1979; Madsen et al., 2006; Melchers, 1999); these define the magnitude of the external force on the base of the probability of exceeding such intensity during the design life of the structure. In addition, the structural safety must be guaranteed on the base of the consequences caused by natural hazards on the structure (vulnerability).

Dealing with natural hazards, one of the common ways to input the external forces applied to the structures is to establish a link between the magnitude of the forces and the corresponding return period. Larger return period implies higher intensity in the force. In a recent work, De Biagi et al. (2016a) have proposed a reliability-based design procedure for structures subjected to snow avalanche hazard.

The magnitude-frequency relationship is at the basis of the probabilistic hazard analysis. In seismic analysis, Gutenberg-Richter's law expresses such relationship. Straub and Schubert (2008) proposed a probabilistic approach for rockfall risk assessment based on a frequency law, but not investigating about its nature. Lari et al. (2014) considered the annual frequency of occurrence of a rockfall volume as a 'given' data. The proposed approach intends to be the base for more complex and complete probabilistic hazard assessments.

In the design of engineering works that must protect a village or a road from falling rocks, e.g., net fences or embankments, at present, the size of the falling block used in the modeling is not linked to its probability of occurrence, i.e., the return period of a block with such volume. The most frequently applied approaches refer to an analysis of the blocks already collapsed integrated with the site surveying on the slope and a choice of the design falling block among them. The adaptation of the well-known procedures to the modern design practice requires that the size of the falling blocks has to be related to its probability of occurrence, and viceversa.

Examples of volume-frequency laws are proposed in the literature. They are obtained from the analysis of a large number of rockfall events for which each observed event is dated and the volume is estimated. This allows to draw a volume frequency curve in which each point corresponds to an observation. In general, precise catalogues with a large number of events are rare because the road owners or the territorial administrations started the records of events, which have large return periods, only some tens of years ago. For common uses, e.g., design of protective devices or risk estimation, for which there are no long records of events nor detailed surveys onsite, no operative procedures are consolidated and the designer develops the project following his personal experience. In any case, the choice of the "characteristic block volume" (design volume) has to be done by designer's own engineering judgement. For this reason, it is affected by subjectivity.

With the aim to contribute to the overcoming of this design problem, this paper proposes a methodology for estimating the block volume frequency relationship that can be used for deriving the size of the design falling block having a prescribed return period. The procedure, which is described in detail in Section 3, is based on the data reported in rockfall inventories and on surveys at the foot of the slope, managed following a statistical procedure.

## 2   Power laws in rockfall analysis

Statistical analysis of historical data or experimental tests related to a certain natural phenomenon gives evidence that it is possible to deduce power laws that link the magnitude of the event to its frequency. These mathematical relationships can be used for predicting type, extent, return time and magnitude of future events.

In the Fifties, Gutenberg and Richter (1956) observed that there was a relationship between cumulative number of earthquake events exceeding a given value of magnitude $N\,(m \geq M)$ and the magnitude itself. They formulated the following law:

$$\log N\,(m \geq M) = \alpha - \beta M \tag{1}$$

where $\alpha$ and $\beta$ are site-dependent constants. More recently, as for earthquakes, statistical analyses of historical data sets have been widely applied to derive the recurrence rate of events of given magnitude for other natural phenomena such landslides, rockfalls, snow avalanches, etc. (Dussauge-Peisser et al., 2002; De Biagi et al., 2012; Corominas et al., 2014).

The analysis of historical data, which are available in public archives or catalogues, is therefore extremely important for the study of natural phenomena. With particular reference to landslides and rockfalls, this statistical approach has been recently studied and applied by several authors in many mountain sites. Research has mainly focused on the analysis of the volume distribution of rockfall events for the sites of Grenoble, Yosemite Valley, Arly gorges, British Columbia, Hong Kong, Italian Apennines, Aosta Valley, Christchurch-Canterbury and La Réunion Island (Dussauge-Peisser et al., 2002; Dussauge et al., 2003; Keith Turner and Schuster, 2012; Abbruzzese et al., 2009; Brunetti et al., 2009; De Biagi et al., 2016b; Guzzetti et al., 1994; Lari et al., 2014).

The comparison of the previous studies showed that negative power laws well fit all rockfall recurrence volume distributions. However, some variability in the values assigned to the power law coefficients does appear. This has been mainly attributed to the variability in the sampling procedures of the landslide volumes. At present, no proper test equipment (which provide, as for earthquakes, objective and reliable values that are comparable from one site to another) and standard procedures have been defined for the different geological and structural settings where rockfalls may occur (Brunetti et al., 2009).

Rockfall inventories do not always contain quantitative and detailed information and the description of historical events is often characterized by a low degree of accuracy. For example, in the Yosemite rockfall inventory (Wieczorek and Snyder, 2004; Guzzetti et al., 2003), which can be considered as one of the largest detailed rockfall inventory, the exact locations where rockfalls occurred, the detachment areas and the block volumes (or weights) are not always given. More often, size and triggering information of the events is given in a qualitative and incomplete way; temporal information is not precise. Rockfalls that occurred within a few hours from the same source area are sometimes listed as the same event, overestimating its magnitude. In general, a lack of data on smaller rock blocks subsists while large and more damaging rockfalls were recorded regularly. Thus, it is clear that a certain degree of uncertainty and lack of homogeneity in the collected data exist.

Previous considerations, which have to be taken into account in treating historical data, are related to Yosemite Valley but can be easily referred to almost all of the historical archives (Corominas et al., 2014; Brunetti et al., 2009; Corominas et al., 2005).

In addition, the temporal length of the observations can affect the recurrence volumetric distribution. In particular, a few years time window underestimates larger collapses. Many authors examined the frequency-size distribution of both rockfalls and fallen blocks and noted that the cumulative frequency is linearly related to the magnitude (block volume or rockfall volume) on a log-log plot. In mathematical terms, the following power law relationship subsists:

$$n(v \geq V) = aV^{-b} \qquad (2)$$

where $n(v \geq V)$ is the frequency of blocks volume (or rockfalls volume) with size larger than $V$ (generally, the size is expressed in $m^3$), while $a$ and $b$ are constants: $a$ relates to the frequency of blocks volume (or rockfalls volume) larger than a unit volume (i.e., 1 $m^3$) and $b$ represents the slope of the regression line, or the fractal dimension (Turcotte, 1997), as sketched in Figure 1. With reference to the example of Figure 1, if the volumes are expressed in cubic meters, $a$ is the annual frequency of occurrence of a rockfall larger than 1 $m^3$. In this case, supposing $V$=2.5 $m^3$, $n(v \geq V)$ is the annual frequency of volumes larger than 2.5 $m^3$.

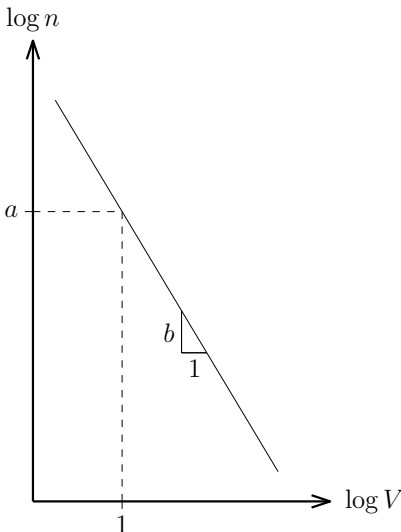

**Figure 1.** Sketch of a $n = aV^{-b}$ power law relationship.

This formulation implies that (i) larger rockfall events are less frequent than those characterized by smaller size and (ii) frequency-size distributions are well fitted by a power law only over a given range of volumes. The power law exhibits a deviation from the observed distribution for volumes smaller than a certain value. This discrepancy has been discussed in the literature. It can be the result of under-sampling of the smallest rockfall events (Brunetti et al., 2009; Stark and Hovius, 2001).

While collapses of considerable sizes are easily identifiable and are almost always recorded, collapses of very small sizes, mainly causing no damage, are unnoticed and, especially in the past, they have been rarely reported in the archives. In addition, the formulation implies that rockfalls of huge sizes can be considered more reliable as much as the recording time increases.

Referring to power laws applied to rockfall volumes, the values of the parameters of Eqn. 2 are variable. Dussauge-Peisser et al. (2002) analyzed a range of volumes spanning from $10^1$ m$^3$ to $10^6$ m$^3$ and suggested that $b$ is not dependent on the scale of

10 study, slope lithology and fracture systems. Other authors propose different values of $b$, depending on the degree of fracturing: the lesser the rock mass fracturing, the smaller the $b$ value. Various studies have been performed for rockfalls less than 10 m$^3$ (Gardner, 1970; Hungr et al., 1999), also by means of topographical techniques down to as $10^{-3}$ m$^3$ (Rosser et al., 2005; Abellán et al., 2010; Dewez et al., 2013). Dai and Lee (2001) studied 2811 landslides and rockfalls and Rousseau (1999) used seismic monitoring technique. On the contrary, coefficient $a$ exhibits relevant fluctuation from one site to another.

As mentioned, Eqn.(2) can be related to the distribution of the volumes of the fallen blocks. The values of the parameters $a$ and $b$ are variable. Parameter $b$ could assume different values in the range 0.5 to 1.3. Various examples can be found in literature. Crosta et al. (2007) determined different fractal dimensions in analyzing grain size curves obtained from different spots of the deposit of a large rock avalanche occurred in 1987 in Central Italian Alps. Ruiz-Carulla et al. (2015) performed a detailed survey in order to highlight the differences in blocks distribution in various portions of the deposit of a rockfall and

found a $b$ value ranging from 0.89 to 1.28. The same authors analyzed the dependency between the free fall height and the value of $b$ for various well documented rockfall events in Spain. They got that $b$ increases as much as the falling height of the blocks increases (Ruiz-Carulla et al., 2016). Observing the data reported in the previously mentioned paper, it emerges that the lithology of the rock mass affects the value of parameter $b$. For similar free fall heights, $b$=0.72 was computed for rockfall in limestones and $b$=0.92 for rockfall in schists. The larger the $b$-value, the more comminuted the deposit. Hantz et al. (2016) surveyed four limestone deposits in the area of Grenoble, France, and found $b$-values ranging from 0.63 to 1.12. Parameter $a$ exhibits relevant variability from one site to another and it is essentially linked to the number of blocks counted on the deposit of the rockfall.

## 3    Proposed method

A three-steps procedure for deriving a volume-frequency relationship for blocks with a reduced number of available data is built up and discussed in the following. Some aspects of the proposed methodology result from hydrological approaches in flood frequency analyses (see Claps and Laio (2003)). The main hypothesis of the procedure is that the temporal occurrences (i.e., the events) are considered separately from the deposit volumes distribution in a representative area where the rockfall phenomenon occurs. A representative area is defined as the portion of deposit beyond a defined line, in which the hazard is computed. We consider the foot of the slope as a representative area.

As described in detail in this section, the required data for deriving a volume-frequency relationship are:

(i) a catalogue of events, i.e., events with quantitative rockfall volume estimates observed in the representative area. The catalogue is denoted as $\mathcal{C}$. Referring to such input, at present, no real-time automatic systems able to detect the occurrence of a rockfall event are diffused. Few examples of monitoring through sensors able to detect microseismic activity are present in the literature. Unfortunately, the calibration of such systems is difficult and the results largely depends on the environmental noises. Other non-real-time methodologies exist. For example, if the phenomena occur in a forested area, the continuous growing of plants can give information about potential impacts (and tree damages) occurred in the past (Dorren et al., 2007). Anyway, this method suffers many epistemic uncertainties: the same rockfall event can damage more than one tree, or, is not possible to distinguish between one or more events occurred during the same plant growing season (Moya et al., 2010). In alternative, topographical approaches, e.g., laser scanning, are largely used to monitor rock faces (Abellán et al., 2010, 2011), but a lasting survey campaign is required to get a robust catalogue of events. The direct observation is still the most common, being a simple and cheap solution for drawing up a catalogue of rockfall events. Usually, local government, road supervisors or forestry service agents are involved in the collection of data related to rockfall events, as reported by Dussauge-Peisser et al. (2002). Since direct observation is affected by errors, in the proposed procedure, a threshold volume is considered, as described in the following;

(ii) a list of measured volumes that may have fallen down at any time. The list is denoted as $\mathcal{F}$. Referring to such input, different counting procedures have been developed. The simplest method consists in counting the fallen blocks and

classifying them into volume classes. Different approaches have been proposed, depending on the size of the rockfall. For example, Corominas et al. (2012) directly counted (and classified) all the fallen blocks in small-size rockfall events occurred in Andorra. For larger phenomena, Ruiz-Carulla et al. (2015) proposed a methodology for obtaining a rockfall block size distribution (RBSD) essentially based on block counting in small sampling plots and homogenization to the whole debris cover. More complex methods make use of topographic techniques (Digital Elevation Models, orthophotos) to identify the existing discontinuity sets and to compute the volume of the unstable rock blocks on the slope face (Jaboyedoff et al., 2009; Mavrouli et al., 2015). In such cases, the time-magnitude relationship would refer to the release of blocks and fragmentation and comminution should be considered in the propagation analysis. In order to avoid this problems, the authors suggest to consider a distribution of volumes obtained from surveys in the representative area.

Obviously, both the catalogue and the list must be related to the same area of the slope, i.e., its foot. All the blocks in catalogue $\mathcal{C}$ are elements of the list $\mathcal{F}$. In addition, the list $\mathcal{F}$ contains also fallen blocks that have not been observed neither recorded. Because of that, its cardinality, i.e., the number of elements, is larger than the one of $\mathcal{C}$.

The first step of the analysis consists in choosing "relevant" events within the catalogue $\mathcal{C}$. To this aim, a threshold volume $V_t$ is identified (details on the choice of $V_t$ are provided in Section 3.1) and the elements of the catalogue $\mathcal{C}$ are split into two sets. The events corresponding to a volume equal or larger than the threshold volume $V_t$ are included in a reduced catalogue $\mathcal{C}^*$ mathematically described as:

$$\mathcal{C}^* = \{e : e \in \mathcal{C} \wedge V(e) \geq V_t\}, \tag{3}$$

where $V(e)$ is the volume associated to falling event $e$. The events not satisfying this condition were discarded and, thus, not considered in the analysis. The list $\mathcal{F}$ is treated in the same way: a list $\mathcal{F}^*$ including all the volumes equal or larger than the threshold volume $V_t$ is set up:

$$\mathcal{F}^* = \{s : s \in \mathcal{F} \wedge V(s) \geq V_t\}, \tag{4}$$

where $V(s)$ is the volume associated to the $s$-th record of the survey at the foot of the slope (in the representative area). As before, the surveyed volumes smaller than $V_t$ are not further considered in the analysis.

The second step of the analysis consists in the choice of two probabilistic models. One should be able to describe the temporal occurrences of the events of catalogue $\mathcal{C}^*$, the other to describe the distribution of the surveyed volumes in the list $\mathcal{F}^*$. It is assumed that the observed events are considered independent if the threshold value, $V_t$, is sufficiently high. Thus, the temporal occurrences can be described with a rare events probabilistic law, i.e., a Poisson distribution. The block sizes at the foot of the slope follow a power law, as previously detailed. A Generalized Pareto Distribution (GPD) is adopted to describe the sizes of the surveyed blocks in the list $\mathcal{F}^*$. The GPD has two degrees of freedom and represents a good compromise between the quality of the fitting (which, in general, increases as much as the number of degrees of freedom increases) and the robustness of the model (which depends on the number of observations). GPD is chosen since it well fits the records of the list $\mathcal{F}^*$, being a power-like distribution, but other probabilistic distributions can be adopted and the proposed procedure easily adapted (Burnham and Anderson, 2003).

Knowing the annual mean number of blocks bigger than $V_t$ (i.e., $\lambda$) and the cumulative distribution function of the block volume $F_V(v)$, the temporal frequency (the inverse of the return period $T$) of blocks bigger than $v$ is:

$$\lambda[1 - F_V(v)] = \frac{1}{T} \tag{5}$$

Inversely, the volume with return period T (vT) is:

$$v_T = F_V^{-1}\left(1 - \frac{1}{\lambda T}\right), \tag{6}$$

where $F_V^{-1}(\cdot)$ is the inverse of the cumulative density function of the probabilistic distribution describing the size of the surveyed blocks, i.e., the Generalized Pareto Distribution, $\lambda$ is the annual mean number of events.

The third step of the analysis consists in the estimation of the parameters of the statistical laws by means of the measured rockfall data contained in $\mathcal{C}^*$ and $\mathcal{F}$. The former gives the parameter temporal frequency, while the latter the parameters of the Generalized Pareto distribution.

## 3.1 Definition of the threshold volume

The catalogue of the events $\mathcal{C}$ contains all the recorded events gathered in a time window, i.e., the beginning and the end of the observation period. For sake of simplicity we consider that the end of the catalogue $\mathcal{C}$ coincides with the present time. The catalogue has a temporal length $\tau(\mathcal{C}) = t$ and is composed by events related to both small and large rockfall phenomena.

Since the recording of the events is related to *in-situ* observations after the occurrence, events involving small rock blocks are not always recorded. Therefore, there is the possibility that the catalogue $\mathcal{C}$ contains only a part of these small events. This fact was considered in the proposed analysis with the introduction of a threshold volume, $V_t$, defined as the minimum size of a fallen block that has always been observed and recorded (after its occurrence). This means that the threshold volume is not necessarily the smallest volume in the catalogue of the events $\mathcal{C}$. This concept is similar to the so called perception threshold in flood frequency analysis (Claps and Laio, 2003).

A reduced catalogue, which is mathematically described by Eqn. (3), is created. The cardinality of $\mathcal{C}^*$, i.e., $|\mathcal{C}^*|$, is equal to $n^*$ and, as already specified, the events are considered independent. The value of the threshold volume influences the temporal length of $\mathcal{C}^*$. Since the decision of monitoring a rockfall prone slope usually begins after the occurrence of an event larger than the threshold volume, it is possible to consider that, in a previous time interval of about half the return period of the events of the reduced catalogue, i.e $t/n^*$, no events were recorded. This means that the temporal length of the reduced catalogue is

$$t^* = \tau(\mathcal{C}^*) = t + \frac{t}{2n^*}. \tag{7}$$

If $n^*$ is larger enough, the term $t/n^*$ is a good estimate of the return period of the events of the reduced catalogue. In the case $n^* = 1$, the return period may be strongly underestimated if the observation period is short.

## 3.2 Probabilistic model describing the temporal occurrence of the events in $\mathcal{C}^*$

Under the hypothesis of independence between the observations, the rockfall phenomenon is considered to be a complete random process for which any realization consists of a set of isolated stochastically independent points in time (McClung,

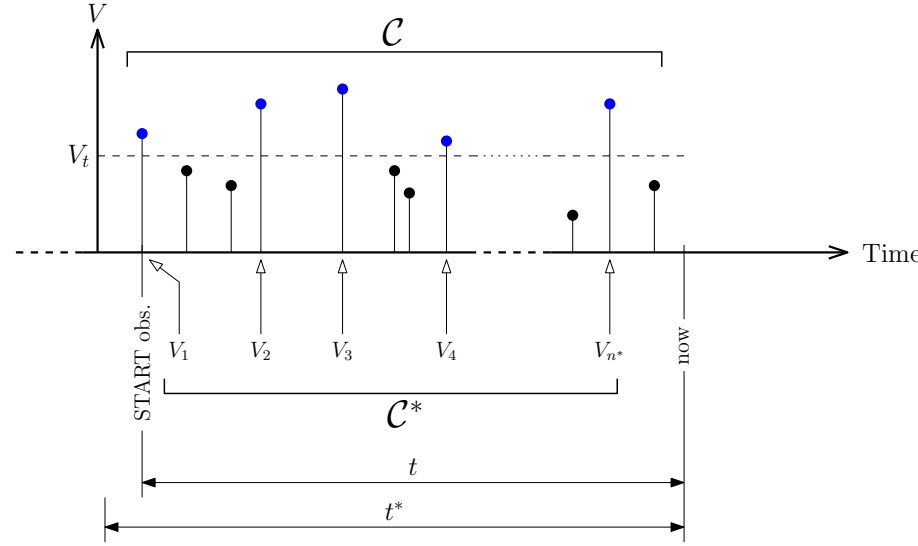

**Figure 2.** Sketch of the catalogues of events $\mathcal{C}$ and $\mathcal{C}^*$. The events which volume $V(e)$ is larger than the threshold volume $V_t$ are indicated with blue bullets, those smaller than the threshold volume $V_t$ are indicated with black bullets.

1999). In statistics, such process is known as Poisson point process. Therefore the events of the reduced catalogue $\mathcal{C}^*$ within the temporal range $t^*$ are considered as a realization of a Poisson point process. The mathematical relationship between the probability of occurrence of $n$ events during the observation period $t^*$, i.e., the probability mass function, is:

$$p(n) = \frac{e^{-\lambda t^*}(\lambda t^*)^n}{n!}, \tag{8}$$

5  where $\lambda > 0$ is the so-called parameter of the Poisson distribution. The hypothesis of independent and Poisson distributed rockfall events is essential to relate the cumulative density function of the sizes of the surveyed blocks, $F_V(v)$, to that of the annual maxima, $G_V(v)$, by means of:

$$G_V(v) = e^{-\lambda[1-F_V(v)]}. \tag{9}$$

$G_V(v)$ represented the annual probability of occurrence of a block of volume smaller than $v$.

10  ### 3.3 Probabilistic model describing the record distribution in $\mathcal{F}^*$

The probabilistic model of the volumes distribution at the foot of the slope is determined using the records contained in the list $\mathcal{F}^*$. As discussed, only blocks larger than the threshold volume, $V_t$, are considered. The Generalized Pareto Distribution (GPD) is used and it has cumulative distribution function equal to:

$$F_V(v) = 1 - \left(1 + \xi \frac{v - \mu}{\sigma}\right)^{-1/\xi}, \tag{10}$$

where $\sigma$, $\xi$ and $\mu$ are scale, shape and location parameters, respectively. The scale parameter is always positive and the distribution has support $v \geq \mu$ for $\xi \geq 0$ or $\mu \leq v \leq \mu - \sigma/\xi$ for $\xi < 0$. The location parameter bounds the distribution. Since the volumes smaller than $V_t$ are not considered, the location parameter is equal to threshold volume, i.e., $\mu = V_t$. The inverse of Eqn.(10), to be used in Eqn.(6), is equal to:

$$v(F_V) = F_V^{-1}(F_V) = \mu + \left[(1 - F_V)^{-\xi} - 1\right] \frac{\sigma}{\xi}. \tag{11}$$

In the present framework, substituting $F_V = 1 - \frac{1}{\lambda T}$, the volume, $v(T)$, corresponding to a return period $T$ years is:

$$v(T) = \mu + \left[(\lambda T)^\xi - 1\right] \frac{\sigma}{\xi} \tag{12}$$

and the return period, $T(v)$, corresponding to a volume $v$ is:

$$T(v) = \frac{1}{\lambda} \left(1 + \xi \frac{v - \mu}{\sigma}\right)^{1/\xi}. \tag{13}$$

By consequence, the annual frequency of occurrence, which is the reciprocal of the return period, is:

$$\frac{1}{T} = \lambda \left(1 + \xi \frac{v - \mu}{\sigma}\right)^{-1/\xi} \tag{14}$$

### 3.4 Evaluation of the parameters of the distributions

An estimate of the parameter $\lambda$ of the Poisson distribution was obtained through the maximum likelihood method. The maximum likelihood estimate is an unbiased estimator of $\lambda$ and was determined as:

$$\lambda = \frac{n^*}{t^*}, \tag{15}$$

i.e., as the ratio between the cardinality and the length of the time window of the catalogue $\mathcal{C}^*$.

While the estimates of the scale and shape parameters, $\xi$ and $\sigma$ respectively, are determined through a maximum likelihood scheme after imposing that the location parameter, $\mu$, is equal to the threshold volume.

## 4 Examples

With the aim to better explain the proposed methodology, it was applied to two areas affected by rockfalls. Both are located in Aosta Valley, Northwestern Italian Alps, as shown in Figure 3: Buisson and Becco dell'Aquila.

### 4.1 Buisson site

Buisson site (UTM: 392267, 5077165, 32, T) is located on the left bank of Marmore torrent in the municipality of Antey-Saint-André in Valtournenche at an altitude ranging from 1130 m to 1612 m a.s.l. The source area is composed of gneiss, which are fine to medium grained rocks with the dominant foliation plane orientation 195/35. Discontinuity sets are observed along 270/85 and 320/80 planes, the latter being the orientation of the slope face. The study slope is mainly composed of debris

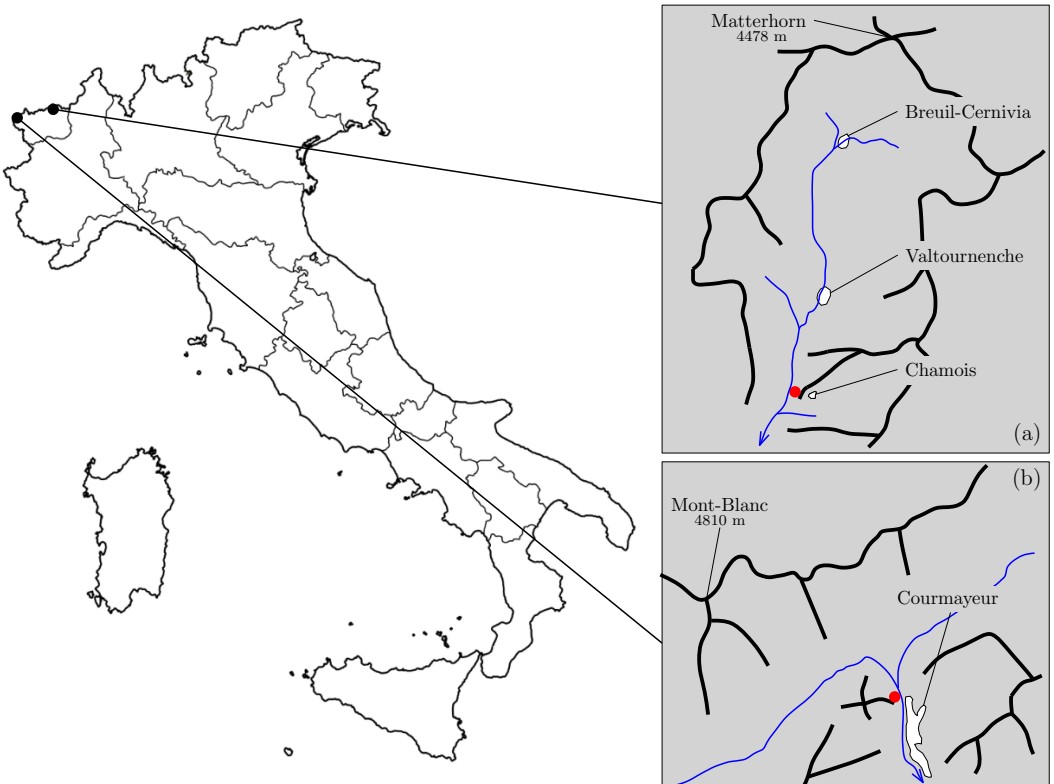

**Figure 3.** Map of the two test sites location in the Northwestern Italian Alps. "Buisson" site is shown with red bullet in subfigure (a), "Becco dell'Aquila" site is shown with red bullet in subfigure (b).

which extends down the slope to the alluvial plain and is covered by irregular (from scanty to very dense) vegetation. The site is close to a camping area, for which protection works and, thus, records of rockfall events, were performed since 1994, when a large block (3 m$^3$) hit a part of the area.

A detailed survey in the deposition area was performed: 60 blocks with volume ranging from 0.02 to 308 m$^3$ were observed and their position recorded through GPS. These data constitute the list $\mathcal{F}$ (Table 1). The analysis of the occurrences was done after the historical catalogue of the Geological Service of Aosta Valley region that reports 5 events in the site from 1994 (Sep 1994, Mar 1995, Sep 1996, Apr 1998, Oct 2002). Because of the continuous monitoring after the construction of the camping, all the events occurred after 1994 were recorded and, thus, considered in the catalogue $\mathcal{C}$ and the reduced catalogue $\mathcal{C}^*$, which are coincident. The threshold volume $V_t$ was set equal to 0.5 m$^3$, i.e., the minimum size of the observed events in $\mathcal{C}$. The number of events considered in the analysis is equal to $n^* = n = 5$. The corrected time $t^*$ is computed through Eqn. (7) and is equal to 25.3 years. Eqn. (15) gives $\lambda = 0.1976$.

| smaller than 0.5 m$^3$ | | | larger than 0.5 m$^3$ | | | |
| --- | --- | --- | --- | --- | --- | --- |
| 0.02 | 0.08 | 0.32 | 0.58 | 1.9 | 8.0 | 58.8 |
| 0.03 | 0.10 | 0.39 | 0.59 | 4.1 | 8.0 | 132.2 |
| 0.03 | 0.11 | 0.40 | 0.66 | 4.1 | 10.0 | 308.6 |
| 0.03 | 0.11 | 0.40 | 1.1 | 4.2 | 11.8 | |
| 0.04 | 0.12 | 0.40 | 1.1 | 4.5 | 18.0 | |
| 0.04 | 0.14 | 0.42 | 1.3 | 4.8 | 18.2 | |
| 0.05 | 0.19 | 0.48 | 1.4 | 5.0 | 18.6 | |
| 0.05 | 0.20 | | 1.6 | 5.1 | 19.2 | |
| 0.07 | 0.21 | | 1.8 | 6.9 | 21.6 | |
| 0.07 | 0.24 | | 1.8 | 7.5 | 24.0 | |

**Table 1.** Volumes of the surveyed blocks in the deposition area of Buisson site. The blocks are divided into two classes, depending on their size (smaller or larger than $V_t$). All the blocks belong to the list $\mathcal{F}$; blocks larger than $V_t$ belong to the reduced list $\mathcal{F}^*$.

The reduced list, $\mathcal{F}^*$, was determined after the definition of the threshold volume, see the right-hand side column of Table 1. The volumes of the reduced list served for the evaluation of the parameters of the Generalized Pareto Distribution, which estimates are reported in the bottom part of Table 2.

| | |
| --- | --- |
| Obs. | 1994-2016 |
| $t$ | 23 yrs |
| $V_t$ | 0.5 m$^3$ |
| $n$ | 5 |
| $n^*$ | 5 |
| $t^*$ | 25.30 yrs |
| $\lambda$ | 0.1976 |
| $\xi$ | 0.994 |
| $\sigma$ | 4.418 |
| $\mu$ | 0.5 |

**Table 2.** Input and results of the analyses performed on Buisson site. The estimates of the parameters of the distribution are reported in the bottom rows.

Figure 4 plots the volume-annual frequency of occurrence relationship given in Eqn. (14). Even if the theory allows the definition of all the possible sizes bigger than $V_t$, an upper threshold value can be introduced taking into account the geostructural surveying of the rock slope that can give evidence of the maximum block size.

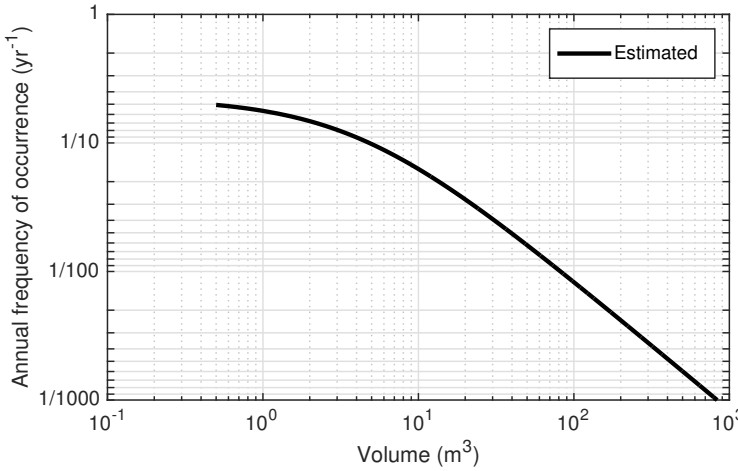

**Figure 4.** Volume-annual frequency of occurrence plot related to Buisson site.

## 4.2 Becco dell'Aquila site

Becco dell'Aquila site (UTM: 341345, 5074157, 32, T) is located on the eastern side of Mont Chétif (2343 m a.s.l.) in the municipality of Courmayeur at an altitude ranging from 1230 m to 1800 m a.s.l. The study area is largely composed of Mont Chétif gneisses, which are fine to medium grained rocks, with the foliation plane N 140/50. The site is close to a deposit
of aggregates used for concrete production onto which a 20 m³ block fell on May 5, 2012. Despite a large rockfall event was recorded in 1903, systematic observations and monitoring activities on the site started in 1998. An onsite survey was performed in the framework of a risk analysis for the activities at the foot of the slope. The block volumes were estimated by rough measurements and through the experience of the geologist. Block sizes are grouped into size classes in a geometric progression following $2^{1/2}$ with volume, as reported in Table 3.
The historical catalogue of the Geological Service of Aosta Valley region reports 3 events in this site since 1998 (Apr 1998, Apr 2001, May 2012). The size of the fallen blocks is always larger than 5 m³. Considering that the slope is constantly monitored it is evident that any event bigger than 5 m³ can be immediately observed and recorded. For this reason a threshold volume $V_t = 5$ m³ was adopted. The catalogue $\mathcal{C}$ and the reduced catalogue $\mathcal{C}^*$ are coincident, see Table 4 for details.

The number of events considered in the analysis is equal to $n^* = n = 3$. The corrected time $t^*$ is computed through Eqn. (7)
and is equal to 22.17 years. Eqn. (15) gives $\lambda = 0.1353$.

Referring to the distribution of the volumes, the reduced list, $\mathcal{F}^*$, was determined after the definition of the threshold volume (Table 3). The neglected data, i.e., belonging to volume classes smaller than 5 m³, are in italic in Table 3. The estimates of the GPD are reported in Table 4.

Based on the previously discussed data it was possible to obtain the volume-annual frequency of occurrence that is reported
in Figure 5 (grey line). A detailed survey of the potential instabilities in the source area showed that the maximum size of the

| Volume (m³) | No. records | Volume (m³) | No. records |
|:---:|:---:|:---:|:---:|
| 1.000 | *10* | 16.000 | 6 |
| 1.414 | *0* | 22.627 | 5 |
| 2.000 | *8* | 32.000 | 2 |
| 2.828 | *0* | 45.255 | 3 |
| 4.000 | *3* | 64.000 | 0 |
| 5.657 | 0 | 90.510 | 1 |
| 8.000 | 26 | 128.000 | 1 |
| 11.314 | 4 | 181.019 | 1 |

**Table 3.** List of the grouped volumes of the surveyed blocks on the slope of Becco dell'Aquila site. All the blocks belongs to the list $\mathcal{F}$; the blocks that are larger than $V_t$, in normal font, belong to the reduced list $\mathcal{F}^*$

| Obs. | 1998-2016 |
|:---:|:---:|
| $t$ | 19 yrs |
| $V_t$ | 5 m³ |
| $n$ | 3 |
| $n^*$ | 3 |
| $t^*$ | 22.17 yrs |
| $\lambda$ | 0.1353 |
| $\xi$ | 0.5509 |
| $\sigma$ | 7.7836 |
| $\mu$ | 5.0 |

**Table 4.** Input and results of the analyses performed on Becco dell'Aquila site. The estimates of the parameters of the distribution are reported in the bottom rows (the standard deviations are detailed into brackets).

detachable block is about 200 m³. Similarly, an additional truncated volume-annual frequency of occurrence relationship is plot (dashed black line).

## 5 Discussions and conclusions

The definition of the relationship between the volumes that can stop on a slope and their return period is a parameter of paramount importance for a correct design procedure. The proposed methodology allows to compute a volume-frequency law that can be used in engineering calculations. Two different probabilistic models are considered: one for the Poisson's point process related to the occurrences of the events, the other for the fallen block volumes distributions (the Generalized Pareto

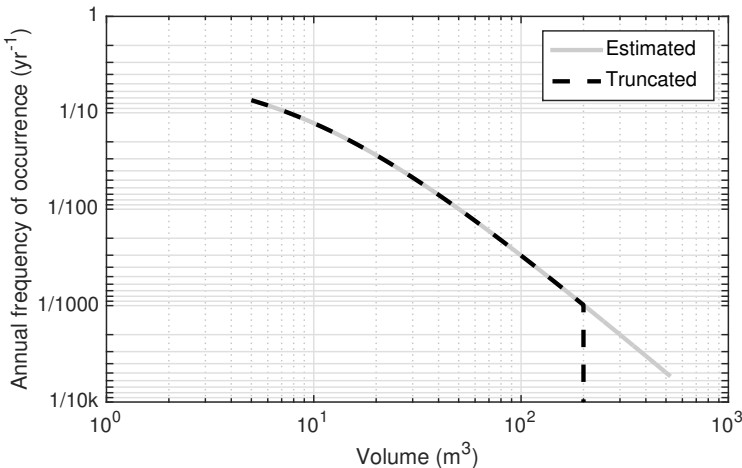

**Figure 5.** Volume-annual frequency of occurrence plot related to Becco dell'Aquila site.

Distribution, which is independent from the rockfall year of occurrence). In order to make these considerations and use these probabilistic models, hypotheses are necessary.

The two probabilistic models are merged considering the hypothesis that the annual frequency of a rock block having volume equal to the threshold volume is the parameter $\lambda$ of the Poisson distribution.

The events described by Poisson's probabilistic models need to be independent. In other words, no causality links have to subsist. Under this hypothesis, the process is random (Moller and Waagepetersen, 2003). In the framework of rockfalls, the validity of hypothesis was discussed by McClung (1999) who stated that the interaction between a natural hazard and anthropic elements (say, vehicles, buildings) is a rare event that can be ascribed to a Poisson process. Similarly, Lari et al. (2014) and Hantz and colleagues (Hantz et al., 2003; Hantz, 2011) invoke the same assumption.

Generalized Pareto Distribution has been chosen for fitting the values of the list $\mathcal{F}$ for various reasons:

- Pareto family distributions are very similar to power law distribution except for the fact that the former are bounded distributions. The bound is represented by the location parameter $\mu$ in Equation (9);

- GPD differs from the classical Pareto model for the introduction of a location parameter, which does not affect the slope of the right part of the plot, being governed by the exponent $-\xi^{-1}$;

- GPD is suitable for extreme value analysis. Pickands (1975) introduced it in the extreme value framework, as the distribution of a sample of exceedances above a certain high threshold.

In rockfall studies, the main distinction between GPD and power law can be observed when the value of the volume tends to zero. GPD is finite for $v \to 0$, while power law diverges to $\infty$, as required by scale invariance (Turcotte, 1997). That is, for the calculations proposed in the present paper, GPD and power law have the same right tail (linear in a log-log plot), while for

small volumes, the former is able to catch the fact that, as much as the volumes are close to the threshold value, $V_t$, a finite number of blocks is counted in the representative area.

The degree of precision of the estimates of the parameters of Generalized Pareto Distribution is determined through a bootstrap analysis (Efron and Tibshirani, 1994; Bengoubou-Valerius and Gibert, 2013). This analysis allows to determine the variance and the confidence bounds of the parameters of the GPD that fit each reduced list $\mathcal{F}^*$ of the previously considered rockfall sites. A hundred thousand bootstrap replications are made for each reduced list. For each replication, a bootstrap sample, i.e., a resampling of the reduced list, is generated and an estimate of the parameters $\xi$ and $\sigma$ of the fitting Generalized Pareto Distribution is made.

From the set of the estimates of the parameters the bootstrap mean, variance and median and the values of 90% and 95% confidence bounds are determined (Table 5). Note that the estimates of the parameters reported in Tables 2 and 4 are close to bootstrap medians.

| | Buisson | | Becco dell'Aquila | |
| --- | --- | --- | --- | --- |
| | $\xi$ | $\sigma$ | $\xi$ | $\sigma$ |
| Mean | 0.937 | 4.919 | 0.511 | 8.104 |
| Variance | 0.166 | 4.511 | 0.032 | 2.226 |
| Median | 0.944 | 4.610 | 0.520 | 7.897 |
| 90% conf.b. | (0.186, 1.572) | (2.247, 8.481) | (0.196, 0.783) | (6.088, 10.871) |
| 95% conf.b. | (0.093, 1.705) | (1.932, 9.563) | (0.103, 0.831) | (5.821, 11.587) |

**Table 5.** Bootstrap statistical parameters of the estimates of the parameters of the Generalized Pareto distribution related to the two example sites.

In addition, for each bootstrap replication, once parameters $\xi$ and $\sigma$ are estimated, the volumes related to different (25 in total) return periods between 10 and 1000 years are computed through Eqn. (12). As an example, the histogram of Figure 6 shows the frequency of the 1000 years return period volumes obtained at Becco dell'Aquila: they are well fit by a lognormal law (red dashed line). The empirical distribution function is plot in the box of Figure 6. The values corresponding to cumulative probability of 0.05 and 0.95 are identified with squares. These corresponds to the bounds of the 90% confidence interval: in other words, this means that 90% of the volumes are larger than 73.8 m$^3$ and smaller than 473.2 m$^3$. Similarly, the bounds are determined on both sites for all the return periods considered in the range 10-1000 yrs. They are reported in dashed in Figure 7.

It results that the width of the 90% confidence interval increases as much as the return period increases. This implies a spread of the value of the volumes of the blocks. Detailed and long records of the occurred events as well as a proper survey of the volumes of the blocks would permit to increase the quality of the volume-frequency law and, by consequence, to reduce the statistical errors in the procedure.

The proposed method allows to define the relationship between the return period and the volume of the blocks. This is a key aspect in land management and planning, design of protection devices (Peila et al., 1998, 2007; Keith Turner and Schuster,

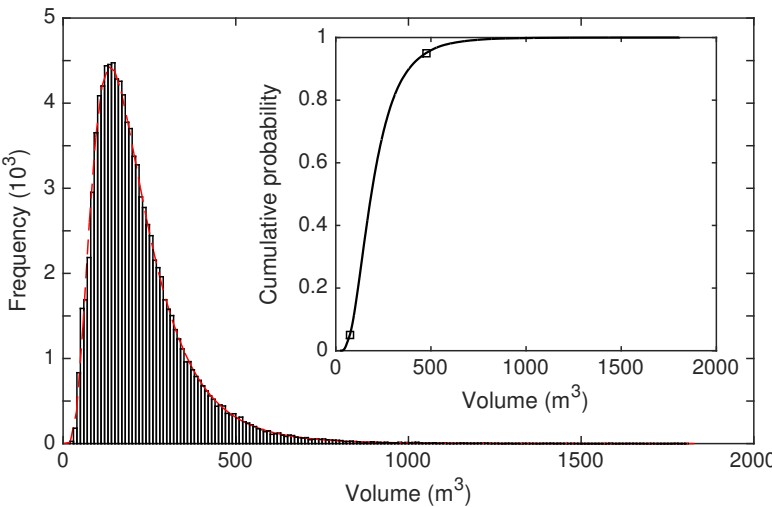

**Figure 6.** Histogram of the volumes having 1000 years return period fitted by a lognormal law (red dashed line). The empirical distribution function is plot in the box: the squares bound the 90% confidence interval.

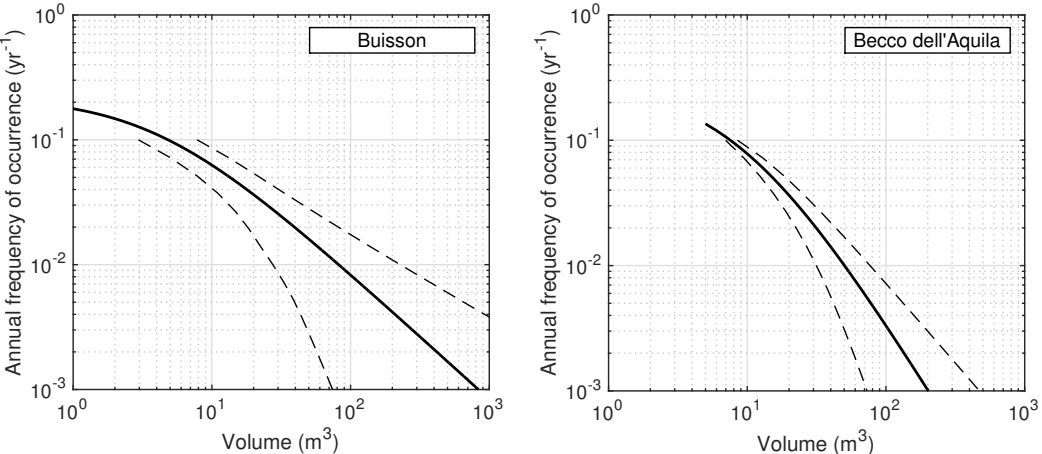

**Figure 7.** Results of the bootstrap analysis on the records on Buisson and Becco dell'Aquila sites. The continuous lines are plot from Eqn. (14) with the parameters reported in Tables 2 and 4. The dashed lines are the bounds of the 90% confidence intervals. In order to easily compare the two, the bounds of the axes are kept equal on the two plots.

2012; Mignelli et al., 2012, 2013; Dimasi et al., 2015) and for for the modern design approaches based on return period of natural hazards (De Biagi et al., 2015, 2016a) and on structural robustness (Cennamo et al., 2015; De Biagi and Chiaia, 2013; De Biagi, 2016). The bootstrap analysis has shown that the quality of the input data can affect the results particularly when long return periods are considered. Hence, in these cases, a critical analysis of the estimated volumes is required in the design

5   process.

### Acknowledgements

The authors acknowledge Dr Hantz and the anonymous referee for their observations and comments of the manuscript. Prof. Laio is really acknowledged for having generously shared his experience and for his valuable comments. This research was supported supported by Regione Autonoma Valle d'Aosta under the framework of the project "Realizzazione di scenari di

10  rischio per crolli di roccia".

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
