# Peer review of "Estimation of the return period of rockfall blocks according to their size"

_Natural Hazards and Earth System Sciences, 2016_

## Referee Comment (RC1) · Anonymous Referee #1 · 25 Oct 2016

The article nicely describes on how to perform a statistical analysis of rockfall samples found in the field in order to obtain a time-and-magnitude-distribution that enables a risk based analysis for the current site.

But even if the procedure provides a final distribution it still depends a lot on the quality of the field survey. The article should therefore bring a little more basics regarding the data aquisition. Which methods do exist and is a certain one recommended? What about the consideration of "modern" methods as e.g. decribed in - Mavrouli, O., Corominas, J., & Jaboyedoff, M. (2015). Size Distribution for Potentially Unstable Rock Masses and In Situ Rock Blocks Using LIDAR-Generated Digital Elevation Models. Rock Mechanics and Rock Engineering, 48(4), 1589-1604.

or more simplified methods such as - Corominas, J., Mavrouli, O., Santana, D., & Moya,

[Figure]

J. (2012). Simplified approach for obtaining the block volume distribution of fragmental rockfalls. Landslides and engineered slopes. Taylor and Francis, 2, 1159-1164.

the volume distributino of the rockfalls strongly depends on the block volumes found in the field. Does the method presented consider that rock blocks often burst into fragments during the rockfall process?

The formulation of a probabilistic distribution of rockfall events based on single samples is also reported in Straub, D., Schubert, M., (2008) Modeling and managing uncertainties in rockfall hazards, Georisk, Assessment and Management of Risk for Engineered Systems and Geohazards, Volume 2, Issue 1, pp. 1-15, DOI: 10.1080/17499510701835696 Maybe, the article can critically compare the method presented there and the actual procedure.

Small typo: P2L26: "e" –> "and"

---

## Referee Comment (RC2) · D. Hantz (Referee) · 28 Oct 2016

The paper deals with the relation between the volume of blocks reaching elements at risk and their temporal frequency (or the return period). This relation is necessary for quantitative rockfall risk assessment and for the design of protection devices. The temporal frequency can be obtained from a catalogue of fallen blocks which have been measured and dated. The distribution of the volumes can be fitted with a probability law. Up to now a power law has been used to describe the distribution of the block volumes. In this paper, a Generalized Pareto Distribution (GPD) is proposed. A common problem for determining the temporal frequency from the observation of fallen blocks is that the fall date of the blocks is unknown (except for the more recent ones). The approach used in this paper consists in considering two catalogues of fallen blocks: recent blocks which have been dated (catalogue C, used for determining the temporal frequency) and

the whole of the observed blocks (catalogue F, used for fitting the Generalized Pareto Distribution).

Comments

Title: The title is not adapted because the paper deals with the return period of blocks and not of rockfalls. I suggest to replace "rockfalls" by "fallen blocks".

1. The section 2 (Power laws in rockfall analysis) is not well adapted because it focuses on studies of rockfall volume distribution (which is not the subject of the paper) instead of block volume distribution (subject of the paper). I suggest references on block volume distribution: Corominas et al., 2005 (already cited); Nocilla et al., 2008 (Rock Mech Rock Eng); Ruiz-Carulla et al., 2015 (already cited), 2016 (Int. Symp. on Landslides); Hantz et al., 2016 (Int. Symp. on Landslides). When reading the paper, it takes a long time before understanding if the analysis concerns rockfall volumes or block volumes. I suggest some corrections in the pdf to clarify this point. The assertion "small rock blocks . . . have been rarely reported in the archives" (page 4, line 1) is true but it must be mentioned here that terrestrial laser scanning allows to build catalogues including very small rockfalls. Examples -Rosser N.J., Petley D.N., Lim M., Dunning S.A., and Allison, R.J.: Terrestrial laser scanning for monitoring the process of hard rock coastal cliff erosion, Q. J. Eng. Geol. Hydrogeol., 38, 363-375, 2005 -Abellan, A., Calvet, J., Vilaplana, J.M., Blanchard, J.: Detection and spatial prediction of rockfalls by means of terrestrial laser scanner monitoring, Geomorphology, 119, 162-171, 2010. - Dewez, T.J.B., Rohmer, J., Regard, V., Cnudde, C. : Probabilistic coastal cliff collapse hazard from repeated terrestrial laser surveys : case study from Mesnil Val (Normandy, northern France), Journal of Coastal Research, 65, 702-707, 2013. The paragraph discussing the values of the exponent b must be rewritten according to the works dealing with the block volume distribution (Ruiz-Carulli et al., 2015, 2016; Hantz et al., 2016; . . .). The values for the rockfall volume distribution are useless in this paper. Particularly, the sentence "the only reliable studies in this range (less than 10 m3) have been performed by Gardner (1970) and Hungr et al. (1999)" must be removed because a lot

of reliable studies have analyzed the volume distribution of smaller rockfalls, down to as 10-3 m3 (for example, Dewez et al. 2013, Journal of Coastal Research).

2. As the hazard (and the risk) is defined for a given point, the Catalogue C should be associated to an element at risk or to a line: Only the blocks which have stopped beyond a defined line should be considered in the analysis. So the notion of "representative area" (page 5, line 2) should be developped.

3. The explanation of Equation (5) is not evident. So I suggest to explain it as follow: Knowing the annual mean number of blocks bigger than Vt ($\lambda$) and the cumulative distribution function of the block volume (FV(v)), the temporal frequency (the inverse of the return period T) of blocks bigger than v is: $\lambda$ (1- FV(v)) = 1/T Inversely, the volume with return period T (vT) is: vT = FV-1(1-1/$\lambda$T) Moreover I suggest to remove the sentence "The combination of the two proposed statistical laws allows to determine the return period . . ." (page 6, line 1), because the Poisson's law is not used (the annual mean number of blocks can be estimated without it).

4. Equation (6) is not evident and should be explained.

5. Section 4 (Examples) As the annual mean number of blocks ($\lambda$) depends of the extent of the considered deposit area, more information should be given (at least the horizontal width and the inclined length of the area). As stated in section 2, the exponent of the power law (and $\lambda$) probably depends on the properties of the rock mass. So the geological and structural context of the Buisson site should be described (rock type and rock mass structure). The orientation of the foliation plane is useless if the orientation of the rock wall is not given (page 11, line 2).

6. As the power law (Equation 2) is commonly used to describe the distribution of the block volume, it should be of interest to compare the volume-annual frequency relations for both Generalized Pareto Distribution (Equation 13) and power law (1/T = $\lambda$ (v/Vt)ˆ-b).
[Figure]

7. Minor corrections are in the pdf.

Please also note the supplement to this comment:
http://www.nat-hazards-earth-syst-sci-discuss.net/nhess-2016-234/nhess-2016-234-RC2-supplement.pdf

**Supplement:**

[revised manuscript text omitted]

---

## Author Comment (AC1) · 7 Nov 2016

The authors acknowledge Anonymous Referee 1 for his interesting comments and observations. First of all, as he underlined, the quality of the time-magnitude distribution depends on the quality of the initial data. In order to highlight the importance of having precise input data, the authors intend to modify the following paragraphs in the manuscript (page 5, line 3).

"*As described in detail in this section, the required data for deriving a volume-frequency relationship are:*

(i) *a catalogue of the observed events, i.e., events with quantitative rockfall volume estimates. The catalogue is denoted as C. Referring such input, at present, no*

*real-time automatic systems able to detect the occurrence of a rockfall event are diffused. Few examples of monitoring through sensors able to detect microseismic activity are present in the literature. Unfortunately, the calibration of such systems is difficult and the results largely depends on the environmental noises. Other non-real-time methodologies exist. For example, if the phenomena occur in a forested area, the continuous growing of plants can give information about potential impacts (and tree damages) occurred in the past (Dorren et al., 2007). Anyway, this method suffers many epistemic uncertainties: the same rockfall event can damage more than one tree, or, is not possible to distinguish between one or more events occurred during the same plant growing season (Moya et al., 2010). In alternative, topographical approaches, e.g., laser scanning, are largely used to monitor rock faces (Abellán et al., 2011), but a lasting survey campaign is required to get a robust catalogue of events. The direct observation is still the most common, being a simple and cheap solution for drawing up a catalogue of rockfall events. Usually, local government, road supervisors or forestry service agents are involved in the collection of data related to rockfall events, as reported by Dussauge et al. (2002). Since direct observation is affected by errors, in the proposed procedure, a threshold volume is considered, as described in the following.*

*(ii) a list of measured volumes that may have fallen down in any time. The list is denoted as F. Referring to such input, different counting procedures have been developed. The simplest method consists in counting the fallen blocks and classifying them into volume classes. Different approaches have been proposed, depending on the size of the rockfall. For example, Corominas et al. (2012) directly counted (and classified) all the fallen blocks in small-size rockfall events occurred in Andorra. For larger phenomena, Ruiz-Carulla et al. (2015) proposed a methodology for obtaining a rockfall block size distribution (RBSD) essentially based on block counting in small sampling plots and homogenization to the whole*

*debris cover. More complex methods make use of topographic techniques (Digital Elevation Models, orthophotos) to identify the existing discontinuity sets and to compute the volume of the unstable rock blocks on the slope face (Jaboyedoff et al., 2009; Mavrouli et al., 2015). In such cases, the time-magnitude relationship would refer to the release of blocks and fragmentation and comminution should be considered in the propagation analysis. In order to avoid this problems, the authors suggest to consider a distribution of volumes obtained from surveys in the representative area.*

*Obviously, both the catalogue. . ."*

Referring to the second observation raised in AR1 comments, the authors propose to add the following paragraph in the Introduction of the manuscript (end of page 1).

"*The magnitude-frequency relationship is at the basis of the probabilistic hazard analysis. In seismic analysis, Gutenberg-Richter's law expresses such relationship. Straub and Schubert (2008) proposed a probabilistic risk approach for rockfall hazard based on a frequency law, but not investigating about its nature. Lari et al. (2014) considered the annual frequency of occurrence of a rockfall volume as a "given" data. The proposed approach intends to be the base for more complex and complete probabilistic hazard assessments.*"

**Additional references**

A. Abellán, J. M. Vilaplana, J. Calvet, D. García-Sellés, and E. Asensio (2011) Rockfall monitoring by Terrestrial Laser Scanning – case study of the basaltic rock face at Castellfollit de la Roca (Catalonia, Spain), Nat. Hazards Earth Syst. Sci. 11: 829–841

J. Corominas, O. Mavrouli, D. Santana, and J. Moya (2012) Simplified approach for obtaining the block volume distribution of fragmental rockfalls. Landslides and Engineered Slopes. Taylor and Francis 2:1159-1164

L. Dorren, F. Berger, M. Jonsson, M. Krautblatter, M. Molk, M. Stoffel, and A. Wehrli

(2007) State of the art in rockfall – forest interactions. Schweizerische Zeitschrift fur Forstwesen 158(6):128-141

C. Dussauge, J. Grasso, and A. Helmstetter (2003) Statistical analysis of rock fall volume distributions: implications for rock fall dynamics. J Geophys Res B 108(B6):2286

M. Jaboyedoff, R. Couture, and P. Locat (2009) Structural analysis of Turtle Mountain (Alberta) using digital elevation model: toward a progressive failure. Geomorphology 103:5-16

S. Lari, P. Frattini, and G.B. Crosta (2014) A probabilistic approach for landslide hazards analysis. Engineering Geology 182:3-14

O. Mavrouli, J. Corominas, and M. Jaboyedoff (2015) Size Distribution for Potentially Unstable Rock Masses and In Situ Rock Blocks Using LIDAR-Generated Digital Elevation Models. Rock Mech Rock Eng 48:1589-1604

J. Moya, J. Corominas, and J. Pérez Arcas (2010) Assessment of the Rockfall Frequency for Hazard Analysis ad Solà d'Andorra (Eastern Pyrenees). In: Tree rings and natural hazards: a State-of-Art, Stoffel et al. (Eds.): 161-176

D. Straub, and M. Schubert (2008) Modeling and managing uncertainties in rockfall hazards. Georisk, Assessment and Management of Risk for Engineered Systems and Geohazards 2:1-15

---

## Author Comment (AC2) · 7 Nov 2016

The authors deeply thank Dr Hantz for his observations and comments to the discussion paper. In the following we answer each point raised in the comment posted on NHESS interactive platform on Oct 28, 2016.

The title will be changed in the final version of the manuscript in accordance to Dr Hantz's comment.

Referring to comment no.1, related to Section 2 (Power laws in rockfall analysis), we will update the manuscript following the suggestions proposed Dr Hantz. In particular, we will clearly state that the focus of the paper is on block volume distribution. This will be first highlighted in the introduction; in particular, we suggest to rewrite the first sentence of the last paragraph of the introduction (page 1, lines 15-16) as "*With the*

[Figure]

*aim to contribute. . .for estimating the block volume frequency relationship that. . .".*

Anyway, in presenting power laws in rockfall analysis we cannot avoid to speak about the studies on rockfall volume distribution. In this sense, we will mention the studies made with laser scanning technique in which very small rockfalls have been recorded (Rosser et al., 2005; Abellan et al., 2010; Dewez et al., 2013).

As suggested, the final paragraph of the section, the one discussing the values of the exponent $b$ of Eqn. (2) will be totally rewritten including the observations made by Dr Hantz. In the following, we propose a possible text:

"*As mentioned, Eqn. (2) can be related to the distribution of the volumes of the fallen blocks. The values of the parameters a and b are variable. Parameter $b$ could assume different values in the range 0.5 to 1.3. Various examples can be found in literature. Crosta et al. (2006) determined different fractal dimensions in analyzing grain size curves obtained from different spots of the deposit of a large rock avalanche occurred in 1987 in Central Italian Alps. Ruiz-Carulla et al. (2015) performed a detailed survey in order to highlight the differences in blocks distribution in various portions of the deposit of a rockfall and found a $b$ value ranging from 0.89 to 1.28. The same authors analyzed the dependency between the free fall height and the value of $b$ for various well documented rockfall events in Spain. They got that $b$ increases as much as the falling height of the blocks increases (Ruiz-Carulla et al., 2016). Observing the data reported in the previously mentioned paper, it emerges that the lithology of the rock mass affects the value of parameter $b$. For similar free fall heights, $b = 0.72$ was computed for rockfall in limestones and $b = 0.92$ for rockfall in schists. The larger the $b$-value, the more comminuted the deposit. Hantz et al. (2016) surveyed four deposits around Grenoble, France, and found $b$-values ranging from 0.63 to 1.12. Parameter $a$ exhibits relevant variability from one site to another and it is essentially linked to the number of blocks counted on the deposit of the rockfall.*"

Referring to comment no.2, the notion of representative area will be better detailed.

In particular, we propose to add in page 5, line 2 the following text: "*A representative area is defined as the portion of deposit beyond a defined line, in which the hazard is computed.*" The need of including in catalogue $\mathcal{C}$ only the events recorded in the representative area will be pointed out at page 5, line 4: "*(i) a catalogue of the observed events in the representative area...*".

Referring to comment no.3, we will update the manuscript according to the suggestions of Dr Hantz, both in the explanation of Eqn. (5) and in the removal of the sentence at page 6, line 1.

Referring to comment no.4, the Equation (6) will be explained as follows (page 8, line 21): "*The value of the threshold volume influences the temporal length of $\mathcal{C}^*$. Since the decision of monitoring a rockfall prone slope usually begins after the occurrence of an event larger than the threshold volume, it is possible to consider that, in a previous time interval of about half the annual mean frequency of the events of the reduced catalogue, i.e $t/n^*$, no events were recorded. This means that the temporal length of the reduced catalogue is*

$$t^* = \tau\left(\mathcal{C}^*\right) = t + \frac{t}{2n^*}. \qquad "$$

Referring to comment no.5, the geological context of the site is described in the following as it will be added to the manuscript (page 8, line 19): "*The source area is composed of gneiss, which are fine to medium grained rocks with the dominant bedding plane orientation 195/35. Discontinuity sets are observed along 270/85 and 320/80 planes, the latter being the orientation of the slope face.*"

Referring to comment no.6, it is important to notice that: "*Generalized Pareto Distribution has been chosen for fitting the values of the list $\mathcal{F}$ for various reasons:*

- *Pareto family distributions are very similar to power law distribution except for the fact that the former are bounded distributions. The bound is represented by the*

*location parameter $\mu$ in Equation (9);*

- *GPD differs from the classical Pareto model for the introduction of a location parameter, which does not affect the slope of the right part of the plot, being governed by the exponent $-1/\xi$;*

- *GPD is suitable for extreme value analysis. Pickands (1975) introduced it in the extreme value framework, as the distribution of a sample of exceedances above a certain high threshold.*

*In rockfall studies, the main distinction between GPD and power law can be observed when the value of the volume tends to zero. GPD is finite for $v \rightarrow 0$, while power law diverges to $\infty$, as required by scale invariance (Turcotte, 1997). That is, for the calculations proposed in the present paper, GPD and power law have the same right tail (linear in a log-log plot), while for small volumes, the former is able to catch the fact that, as much as the volumes are close to the threshold value, $V_t$, a finite number of blocks is counted in the representative area.*" The previous text will be inserted in page 13, line 6.

**Additional references** A. Abellán, J. Calvet, J.M. Vilaplana, and J. Blanchard (2010) Detection and spatial prediction of rockfalls by means of terrestrial laser scanner monitoring. Geomorphology 119:162-171.

G.B. Crosta, P.Frattini, and N. Fusi (2007) Fragmentation in the Val Pola rock avalanche, Italian Alps. J. of Geophysical Res. 112:F01006

T.J.B. Dewez, J. Rohmer, V. Regard, and C. Cnudde (2013) Probabilistic coastal cliff collapse hazard from repeated terrestrial laser surveys : case study from Mesnil Val (Normandy, northern France). Journal of Coastal Research 65:702-707

D. Hantz, Q. Ventroux, J.P. Rossetti, and F. Berger (2016) A new approach of diffuse rockfall hazard. In: Aversa, Cascini, Picarelli and Scavia (Eds.) Landslides and Engineered Slopes. Experience, Theory and Practice: Proceedings of the 12th International Symposium on Landslides (Napoli, Italy, 12-19 June 2016), 2:1063-1068

J. Pickands (1975) Statistical inference using extreme order statistics. Ann. Statist. 3:119–131

N.J. Rosser, D.N. Petley, M. Lim, S.A. Dunning, and R.J. Allison (2005) Terrestrial laser scanning for monitoring the process of hard rock coastal cliff erosion. Q. J. Eng. Geol. Hydrogeol. 38:363-375

R. Ruiz-Carulla, J. Corominas, and O. Mavrouli (2016) Comparison of block size distributions in rockfall. In: Aversa, Cascini, Picarelli and Scavia (Eds.) Landslides and Engineered Slopes. Experience, Theory and Practice: Proceedings of the 12th International Symposium on Landslides (Napoli, Italy, 12-19 June 2016), 3:1767-1774
* * *
**NHESSD**

---

## Author Response (AR1)

Dear Editor,

first of all we gratefully thank all the people involved in the revision of our manuscript, in particular Anonymous Referee no.1 and Dr Hantz (Referee no.2) for their suggestions and corrections to the manuscript which certainly improved its overall quality.

Following the questions of Anonymous referee no.1 and Dr Hantz, some paragraphs and appropriate references have been added to the manuscript. In addition, English language minor errors have been corrected properly.

The manuscript has been updated with the corrections and the remarks on the language that the two reviewers made. In the following version of the manuscript in which the relevant changes are highlighted, the modifications related to the observations and comments made by Anonymous Referee no.1 are in red, those related to Referee no.2 (Dr Hantz) are in blue. In the following "Authors' response" letter, the comments of each referee are in bold, while authors' reply is in normal character.

We hope that our revision has fulfilled editor's and reviewers' requests.

Yours truly,

Valerio De Biagi (corresponding author)
Maria Lia Napoli
Monica Barbero
Daniele Peila

Anonymous referee no.1

In the "interactive comment" published on the online procedure on Oct 25, 2016, anonymous referee no.1 points out few observations and questions:

**AR1: But even if the procedure provides a final distribution it still depends a lot on the quality of the field survey. The article should therefore bring a little more basics regarding the data acquisition. Which methods do exist and is a certain one recommended? What about the consideration of "modern" methods as e.g. decribed in - Mavrouli, O., Corominas, J., & Jaboyedoff, M. (2015). Size Distribution for Potentially Unstable Rock Masses and In Situ Rock Blocks Using LIDAR-Generated Digital Elevation Models. Rock Mechanics and Rock Engineering, 48(4), 1589-1604. or more simplified methods such as - Corominas, J., Mavrouli, O., Santana, D., & Moya, J. (2012). Simplified approach for obtaining the block volume distribution of fragmental rockfalls. Landslides and engineered slopes. Taylor and Francis, 2, 1159-1164.**

AUTH: Basics on data acquisition have been provided in the manuscript. In particular, referring to the catalogue of the events, the various approaches suggested in the literature have been reported (page 5, lines 19 to 31). Referring to the distribution of the measured volumes, the basic survey techniques have been added to the manuscript (from page 5, line 32 to page 6, line 7).

**AR1: The volume distribution of the rockfalls strongly depends on the block volumes found in the field. Does the method presented consider that rock blocks often burst into fragments during the rockfall process?**

AUTH: The present analysis deals with the distribution of the values of the fallen blocks volumes. In this sense, a sentence has been added clarifying this point and the need of having a comminution/fragmentation model to correlate rockfall volume to blocks volume (page 6, lines 7 to 9).

**AR1: The formulation of a probabilistic distribution of rockfall events based on single samples is also reported in Straub, D., Schubert, M., (2008) Modeling and man- aging uncertainties in rockfall hazards, Georisk, Assessment and Management of Risk for Engineered Systems and Geohazards, Volume 2, Issue 1, pp. 1-15, DOI: 10.1080/17499510701835696 Maybe, the article can critically compare the method presented there and the actual procedure.**

AUTH: The connection of the procedure presented in our manuscript to other risk assessments found in the literature has been pointed out in the "Introduction" section, in particular at page 2, lines 1 to 5.

In the "Interactive comment" and supplement PDF published on the online procedure on Oct 28, 2016, Dr Hantz points out few observations and questions:

**R2: Title: The title is not adapted because the paper deals with the return period of blocks and not of rockfalls. I suggest to replace "rockfalls" by "fallen blocks".**
AUTH: The title of the manuscript has been changed according to the observation of Referee no.2

**R2: The section 2 (Power laws in rockfall analysis) is not well adapted because it focuses on studies of rockfall volume distribution (which is not the subject of the paper) instead of block volume distribution (subject of the paper). I suggest references on block volume distribution: Corominas et al., 2005 (already cited); Nocilla et al., 2008 (Rock Mech Rock Eng); Ruiz-Carulla et al., 2015 (already cited), 2016 (Int. Symp. on Landslides); Hantz et al., 2016 (Int. Symp. on Landslides). When reading the paper, it takes a long time before understanding if the analysis concerns rockfall volumes or block volumes. I suggest some corrections in the pdf to clarify this point. The assertion "small rock blocks…have been rarely reported in the archives" (page 4, line 1) is true but it must be mentioned here that terrestrial laser scanning allows to build catalogues including very small rockfalls. Examples -Rosser N.J., Petley D.N., Lim M., Dunning S.A., and Allison, R.J.: Terrestrial laser scanning for monitoring the process of hard rock coastal cliff erosion, Q. J. Eng. Geol. Hydrogeol., 38, 363-375, 2005 -Abellan, A., Calvet, J., Vilaplana, J.M., Blanchard, J.: Detection and spatial prediction of rockfalls by means of terrestrial laser scanner monitoring, Geomorphology, 119, 162-171, 2010. - Dewez, T.J.B., Rohmer, J., Regard, V., Cnudde, C. : Probabilistic coastal cliff collapse hazard from repeated terrestrial laser surveys : case study from Mesnil Val (Normandy, northern France), Journal of Coastal Research, 65, 702-707, 2013. The paragraph discussing the values of the exponent b must be rewritten according to the works dealing with the block volume distribution (Ruiz-Carulli et al., 2015, 2016; Hantz et al., 2016;…). The values for the rockfall volume distribution are useless in this paper. Particularly, the sentence "the only reliable studies in this range (less than 10 m3) have been performed by Gardner (1970) and Hungr et al. (1999)" must be removed because a lot of reliable studies have analyzed the volume distribution of smaller rockfalls, down to as 10-3 m3 (for example, Dewez et al. 2013, Journal of Coastal Research).**
AUTH: According to the observations of Referee no.2, a proper paragraph dealing with the values of exponent "b" in the power laws related to blocks volumes distribution has been inserted (from page 4, line 16 to page 5, line 9). In parallel, the manuscript dealing with exponent "b" in the power laws related to rockfall volumes distribution has been rewritten following the suggestions of Dr Hantz: in particular, the values of "b" found in other studies on rockfall volumes distribution have not been reported. The paragraph, i.e., page 4, from line 9 to line 15, serves for completeness in the treatise of power laws in rockfall studies.

**R2: As the hazard (and the risk) is defined for a given point, the Catalogue C should be associated to an element at risk or to a line: Only the blocks which have stopped beyond a defined line should be considered in the analysis. So the notion of "representative area"**

**(page 5, line 2) should be developped.**

AUTH: The notion of representative area has been clearly detailed. In particular, a sentence has been added (page 5, lines 15 and 16) in the appropriate paragraph.

**R2: The explanation of Equation (5) is not evident. So I suggest to explain it as follow: Knowing the annual mean number of blocks bigger than Vt () and the cumulative distribution function of the block volume (FV(v)), the temporal frequency (the inverse of the return period T) of blocks bigger than v is: (1- FV(v)) = 1/T Inversely, the volume with return period T (vT) is: vT = FV-1(1-1/T) Moreover I suggest to remove the sentence "The combination of the two proposed statistical laws allows to determine the return period . . ." (page 6, line 1), because the Poisson's law is not used (the annual mean number of blocks can be estimated without it).**

AUTH: The manuscript has been rewritten according to the observations of Referee no.2 (page 7, lines 1 to 4).

**R2: Equation (6) is not evident and should be explained.**

AUTH: The explanation of Equation (6), which is labelled as (7) in the updated version of the manuscript, has been reported in page 7, lines 23 to 26. In order to better explain the temporal extension of the observation interval, Figure 2 has been slightly modified showing that the first event with volume larger than threshold volume $V_t$ occurs when the observation period starts.

**R2: Section 4 (Examples) As the annual mean number of blocks () depends of the extent of the considered deposit area, more information should be given (at least the horizontal width and the inclined length of the area). As stated in section 2, the exponent of the power law (and) probably depends on the properties of the rock mass. So the geological and structural context of the Buisson site should be described (rock type and rock mass structure). The orientation of the foliation plane is useless if the orientation of the rock wall is not given (page 11, line 2).**

AUTH: Geological details on bedding plane orientation, discontinuity sets and slope face orientations have been added in page 9, lines 24 to 26.

**R2: As the power law (Equation 2) is commonly used to describe the distribution of the block volume, it should be of interest to compare the volume-annual frequency relations for both Generalized Pareto Distribution (Equation 13) and power law (1/T = (v/Vt)ˆ-b).**

AUTH: The choice of using Generalized Pareto Distribution as well as its similarities with power law distribution have been discussed in "Discussion and conclusion" section, in particular from page 14, line 11 to page 5, line 3.

**R2: Minor corrections are in the pdf.**

AUTH: The manuscript has been modified accordingly.

[revised manuscript text omitted]

---

## Author Response (AR2)

Dear Editor,

first of all we gratefully thank all the people involved in the revision of our manuscript, in particular Dr Hantz for his suggestions and corrections to the manuscript which certainly improved its overall quality. They have been acknowledged in the appropriate section of the manuscript.

Following the observations made by Dr Hantz, the manuscript has been corrected.

In the following "Editor' response" letter, the comments of Dr Hantz are in bold, while authors' reply is in normal character.

We hope that our revision has fulfilled editor's and reviewer's requests.

Yours truly,

Valerio De Biagi (corresponding author)
Maria Lia Napoli
Monica Barbero
Daniele Peila

In the "Report #1" published on the online procedure on Dec 9, 2016, Dr Hantz points out few observations:

**DrH: The most important remarks and comments have been taken into account in the revised manuscript, but some minor revisions are still necessary. They are listed below. Moreover, some minor corrections suggested in the pdf of the first version are still necessary.**
AUTH: The minor corrections suggested in the PDF of the first version have been made.

**DrH: 1. In their interactive comment, the authors intend to mention the studies of Rosser et al. (2005), Abellan et al. (2010) and Dewez et al. (2013), but they mention only Dewez et al. in the new manuscript (line 13, page 4). Rosser et al. are also forgotten in the reference list.**
AUTH: The references have been added to the manuscript (page 4, line 13).

**DrH: 2. At the end of section 2 (page 5), when discussing the influence of the lithology, it should be mentioned that the deposits surveyed by Hantz et al. (2016) consists is a limestone deposit (page 5, line 7).**
AUTH: This information has been added "Hantz et al. (2016) surveyed four limestone deposits in…" (page 5, line 5-6).

**DrH: 3. Page 9, line 25, the term bedding should be replaced by foliation, which is convenient for metamorphic rocks.**
AUTH: The term has been replaced (page 9, line 25).

**DrH: An error in Equation (5) must be corrected (lambda is missing):**
**Replace "(1- FV(v)) = 1/T" by "lambda (1- FV(v)) = 1/T".**
AUTH: The equation has been corrected (Page 7, Eqn.(5)).

**DrH: Page 7, line 25 and 28, "half the annual mean frequency" must be replaced by "half the return period". Moreover, it should be mentioned that t/n\* is a good estimate of the return period only if n\* is high enough (with n\* = 1, the return period may be strongly underestimated if the observed period is short!).**
AUTH: "Half the annual mean frequency" was replaced by "half the return period" in page 7, line 24. In addition, as suggested by Dr Hantz, page 7, lines 27-28 have been rewritten accordingly: "If $n^*$ is larger enough, the term $t/n^*$ is a good estimate of the return period of the events of the reduced catalogue. In the case $n^* = 1$, the return period may be strongly underestimated if the observation period is short."

**DrH: Page 8, line 13: Equation (10) isn't the cumulative "density" function but the cumulative "distribution" function.**
AUTH: The term has been corrected (page 8, line 13).

**DrH: Page 12, line 7: "the block" is the beginning of a new sentence.**
AUTH: The error has been corrected (page 12, line 7).

**DrH: Page 12, line 4: Replace "with the dominant bedding planes orientation of the foliation N140/50" by "with the foliation plane N140/50".**
AUTH: The sentence has been replaced (page 12, line 4).

**DrH: Page 13, line 5: "detach from" should be replaced by "stop on", because all the blocks which detached from the slope are not surveyed, but only those which have reached the observed area.**
AUTH: The term has been replaced (page 13, line 4).

**DrH: Page 14, line 9, replace "colleauges" by "colleagues"**
AUTH: The error has been corrected (page 14, line 9).